# A Solvent-Mediated Excited-State Intermolecular Proton Transfer Fluorescent Probe for Fe^3+^ Sensing and Cell Imaging

**DOI:** 10.3390/molecules27020516

**Published:** 2022-01-14

**Authors:** You Qian, Fuchun Gong, Jiguang Li, Pan Ma, Hanming Zhu, Lingzhi He, Jiaoyun Xia

**Affiliations:** 1College of Chemistry and Chemical Engineering, Changsha University of Science and Technology, Changsha 410114, China; qianyou777@126.com (Y.Q.); jwcjyk2004@163.com (P.M.); zhuhanming666@126.com (H.Z.); helingzhi@sioc.ac.cn (L.H.); jiaoyuuxia@126.com (J.X.); 2Crop Research Institute, Hunan Academy of Agricultural Sciences, Changsha 410125, China; lzh_cs@csust.edu.cn

**Keywords:** solvent-mediated ESIPT fluorophor, tricolour emission, Fe^3+^ sensing, cell-imaging

## Abstract

Constructing excited-state intermolecular proton transfer (ESIPT-e) fluorophores represents significant challenges due to the harsh requirement of bearing a proton donor-acceptor (D-A) system and their matching proton donating-accepting ability in the same molecule. Herein, we synthesized a new-type ESIPT-e fluorophor (2-APC) using the “four-component one-pot” reaction. By the installing of a cyano-group on pyridine scaffold, the proton donating ability of -NH_2_ was greatly enhanced, enabling 2-APC to undergo ESIPT-e process. Surprisingly, 2-APC exhibited dual-emissions in protic solvents ethanol and normal fluorescence in aprotic solvents, which is vastly different from that of conventional ESIPT-a dyes. The ESIPT emission can be obviously suppressed by Fe^3+^ due to the coordination reaction of Fe^3+^ with the A-D system in 2-APC. From this basis, a highly sensitive and selective method was established using 2-APC as a fluorescent probe, which offers the sensitive detection of Fe^3+^ ranging from 0 to 13 μM with the detection limit of 7.5 nM. The recovery study of spiked Fe^3+^ measured by the probe showed satisfactory results (97.2103.4%) with the reasonable RSD ranging from 3.1 to 3.8%. Moreover, 2-APC can also exhibit aggregation-induced effect in poor solvent or solid-state, eliciting strong red fluorescence. 2-APC was also applied to cell-imaging, exhibiting good cell-permeability, biocompatibility and color rendering. This multi-mode emission of 2-APC is significant departure from that of conventional extended p-conjugated systems and ESIPT dyes based on a flat and rigid molecular design. The “one-pot synthesis” strategy for the construction of ESIPT molecules pioneered a new route to achieve tricolor-emissive fluorophores.

## 1. Introduction

The proton transfer reaction is one of the most fundamental and important photophysical and photochemical processes in chemistry and biology [1,2]. Excited-state proton transfers have been investigated extensively because the reactions can be triggered easily by photoexcitation and elicit interesting photophysical and photochemical change [3,4]. The excited-state intramolecular proton transfer (ESIPT-a) have been acquired considerable attention, in which the proton donor and acceptor sites exist in the same molecule and form a cyclic intramolecular hydrogen bonded (IMHB) ring in the ground state. Most of the ESIPT-a reaction occurs in the same molecule bearing IMHB ring in the ground state, in which the intrinsic proton transfer from the enol to keto form in the π-π* state is barrierless in gas-phase and aprotic solvents [5,6,7]. In 1956, the first observation of ESIPT-a fluorescence evidence was reported by Weller and co-workers [8]. Since then, theoretical and experimental research into the ESIPT-a process has received considerable attention from scholars because of its potential application such as laser dyes [9,10], photo-stabilizers [11,12], fluorescent probes for bioimaging [13,14] and light-emitting devices [15,16]. However, as the formation of an IMHB ring is a prerequisite for ESIPT reaction, the ESIPT-a fluorophores have been limited to only a few variations, such as planar molecules that consist of a phenol or an aniline scaffold as a proton donor (D) and an imino/azo-nitrogen or carbonyl-oxygen containing moiety as a proton acceptor (A). Moreover, the ESIPT-a reaction usually takes place in aprotic solvents or a hydrophobic environment, which limits their practical application, especially in biological systems.

Another kind of ESIPT reaction is usually called the excited-state intermolecular proton transfer (ESIPT-e) process. Similar to the ESIPT-a, this category includes the system having a D and an A site, but the D-A systems involved in the proton transfer are far from each other or do not have the proper geometry to form an IMHB ring [17,18]. They can undergo the ESIPT-e process only if they partner with other molecules with both H-bond accepting and donating abilities [19,20]. For example, the double proton transfer often occurs via a concerted process from one functional group to another, as was reported in the case of double H-bonded dimers of 7-azaindole [21,22] or relayed by a cyclic bridge of protic solvent molecules (methanol, ethanol and ammonia, etc.) between two different groups of the partnered molecules such as in 7-hydroxyquinoline (7-HQ) [23,24]. In the partnering systems, the formation of H-bonds between their siblings or between the ESIPT-e host and solvent molecule in the ground state can facilitate proton transfer in the excited state and show ESIPT-e emission [25,26]. In contrast to ESIPT-a species, ESIPT-e systems can only undergo the ESIPT reaction based on partnering with protic solvent or homologous molecules [27,28]. The ESIPT-e reaction in nitrogen-containing heterocyclic compounds is particularly relevant to biochemical systems, which may enable them to become outstanding biological probes, especially for detecting DNA mutations and the function of vitamin B1 coenzyme [29,30]. However, like the ESIPT-a system, the ESIPT-e molecules must possess a D-A system and matching proton donating-accepting capacity in the same molecules, implying that this harsh requirement poses a great challenge to the design and synthesis of ESIPT-e compounds. Up to now, the studies on ESIPT-e compounds is still limited to the experimental observation and theoretical estimation of several simple molecules.

Among ESIPT-e molecules, 2-aminopyridine (2-AP) as a typical amino-imino D-A system, has drawn much attention of several groups [31,32,33]. It has been reported that the excited-state double proton transfer (ESDPT) can take place in the 2-AP/acetic acid system because of the formation of eight-membered intermolecular double H-bonds by partnering with acetic acid molecules. However, the 2-AP cannot occur in the ESIPT-e reaction in protic solvents except for acetic acid due to the insufficient acidity and poor proton donating ability of -NH_2_ [34,35].

Fe^3+^ is one of the most abundant transition metals in biosystems and participates in important biological functions, such as oxygen uptake, oxygen metabolism, electron transfer, and transcriptional regulation [36,37]. Fe^3+^ deficiency and overload may cause various diseases, including Parkinson’s disease, Alzheimer’s disease and cancer [38,39]. Therefore, the development of Fe^3+^-specific probes, as efficient and economic methods for the analysis of Fe^3+^, is very important in biological systems and environmental science.

In order to obtain a sensitive, selective and simple method for the detection of Fe^3+^, we designed a new type of ESIPT-dye (2-APC) using 2-AP as an ESIPT scaffold, which bears a proton A-D system and an auxochrome. We found that this architecture can undergo ESIPT reaction smoothly in conventional protic solvents, eliciting the unique characteristics of dual-emission and aggregation-induced emission. In the use of 2-APC as a fluorescent probe, a sensitive method for Fe^3+^ detection and cell-imaging was developed.

## 2. Results and Discussion

### 2.1. Synthesis and Characterization of 2-APC

Considering the criteria of the ESIPT reaction, we employed 2-aminopyridine (2-AP) as an ESIPT scaffold and installed an electron-withdrawing cyano group (-CN) on the 2-AP ring to increase the proton donating acidity of -NH_2_. By incorporating a rigid and large conjugated aromatic n-ethylcarbazole with the cyano-modified 2-AP, a new architecture bearing a D-A system and an auxochrome can be obtained. We hope to facilitate the proton transfer reaction of 2-AP-based ESIPT molecules and improve their fluorescence properties through this design. As outlined in Figure 1 the target compound 2-amino-4-(9-ethyl-9H-carbazol-3-yl)-6-phenylnicotinonitrile (2-APC) was readily achieved by the “four-component one-pot” reaction of N-ethylcarbazolecarboxaldehyde, malonitrile, acetophenone and ammonium acetate at refluxing temperature in ethanol. The as-prepared compounds were easily purified by column chromatography or recrystallization with a reasonable yield. 2-APC was fully characterized by spectroscopic methods, such as 1H NMR, 13C NMR and Fourier transform infrared spectroscopy (FT-IR), with satisfactory results (in Appendix A).

### 2.2. Optical Properties of 2-APC

The absorption and fluorescence spectra of 2-APC in aprotic solvents and protic solvents were measured. As shown in Figure 1A, the absorption peak located at around 335 nm is pointed to the π-π* transition. When the 6-position on pyridine ring was occupied by aryl group, 2-APC exhibited the longer absorption band at 446 nm, which may be attributed to the intramolecular charge transfer (ICT) transition from the electron-donating group to the electron-accepting 6-substituent [40,41]. The fluorescence profiles showed that 2-APC can exhibit only one emission peak at 443 nm in dioaxne. Upon the addition of ethanol or water, two emission peaks at 443 and 550 nm can be observed (Figure 1B, C). 2-APC can exhibit tricolor emission, in which the fluorescence in dixoane (aprotic solvent) is vastly different from that in ethanol and solid-state (Figure 1D). It is clear that 2-APC can undergo the ESIPT reaction in a protic solvent, resulting in a dual-emission. The emission characteristics of 2-APC in ethanol–water was also investigated. As shown in Figure 2A, the fluorescence spectra show distinct red-shifting upon change in the percentage of water content to the ethanol solution. In a pure ethanol solution of 2-APC, the fluorescence bands appear at 376 and 547 nm, and upon the addition of water to this solution, a new band appears at 595 nm, which is an indication of J-aggregates [42]. The fluorescence microscopic image also shows that there are J-aggregates of 2-APC in ethanol–water mixtures (Figure 2B). The AIE effect may be because of the high rigidity of the probe in an aggregated state, which may cause the restriction of intramolecular rotation due to the H-bonding.

The time-resolved fluorescence of 2-APC in dioxane, ethanol and the ethanol–water mixtures, as well as in solid-state, was also monitored. As shown in Figure 3, the fluorescence decay of 2-APC in dioxane measured at 443 nm is very fast (=0.4382 ns). When the time-resolved fluorescence of 2-APC was tested at 527 nm in ethanol and ethanol–water mixtures, their decay rates are almost the same (ethanol: 1.36 ns; ethanol–water: 1.51 ns), corresponding to the ESIPT reaction time. However, the fluorescence lifetime of 2-APC in solid-state is 4.47 ns, which implies other photophysical process. The time-resolved fluorescence measurement reveals three lifetimes for the peaks of normal, ESIPT and AIE in varying solvents, as well as in solid-state, which indicates that 2-APC can show tricolor-emission based on multi-mechanisms.

### 2.3. Comparative Studies on the Net Charge Distributions of D-A Systems

To decipher the detailed mechanism of ESIPT reaction of 2-APC, the net charge distributions of the proton donor and acceptor in 2-amino-pyridine, 2-amino-3-cyanopyridine and 2-APC were comparatively studied using the Extended Hückel method. As shown in Table 1, the net charges of pyridine-N (A) and NH_2_ (D) in 2-amino-pyridine molecules are −0.398859 and 0.3203749, respectively. When the cyano-group is grafted to the *o*-position of NH2, the value of A and D on 2-amino-3-cyanopyridine become −0.381719 and 0.3453929, respectively. The net charge of NH2 increases only to a certain extent, which implies that cyano-substitution on aromatic molecules may possibly lead to the dramatic change in acidity, agreeing with those reported in the literature [43,44]. In our newly developed 2-APC molecule, the net charge values of A and D are −0.457820 and 0.3403941, respectively. Compared with that of 2-amino-3-cyanopyridine, it is obvious that both the acidity of D and the basicity of A on 2-APC are increased apparently, implying an enhanced proton transfer tendency in 2-APC molecules.

Based on the theoretical estimation and the experimental evidence of reported results [45,46], the corresponding theoretical interpretation of the possible mechanisms of partnered excited-state proton transfer and the subsequent tricolor-emissions was proposed as shown in Figure 2. In aprotic solvents, 2-APC exhibits only a single fluorescence band. When 2-APC is present in the protic solvents, the 2-APC molecules can partner with the solvent molecules to form a cyclic 2-APC/solvent hydrogen-bonded complex and undergo ESIPT reaction, eliciting dual-emission. In solid-state or concentrated solution, 2-APC can not only partner with their compatriots and enable the ESIPT process but also associate with their excited siblings to create dimers, resulting in aggregation-induced emission.

### 2.4. Sensing Behavior of 2-APC to Fe^3+^

The sensing performance of 2-APC towards Fe^3+^ was examined in an ethanol–HEPES buffer (10 mM, pH 6.5) mixed system at room temperature. As shown in Figure 1A, when the Fe^3+^ solution was added into the 2-APC-containing solution, the color turned from light yellow to brown, accompanied by a definite change in absorption spectra. The fluorescence spectrum of 2-APC indicates two emission peaks in an ethanol–HEPES system, representing a typical ESIPT emission. Upon the excitation of 345 nm UV-light, strong ESIPT fluorescence of 2-APC can be obtained. The addition of 2.5 μM Fe^3+^ elicits obvious quenching of the ESIPT fluorescence of 2-APC (Figure 1C). Meanwhile, a distinct change in the bright green fluorescence of the 2-APC solution can be observed under 365 nm UV light (Figure 3D). From this basis, we established a sensitive and selective method for detecting Fe^3+^ and cell-imaging using 2-APC as a probe.

The principle of this signaling system based on ESIPT emission quenching of 2-APC is suggested. There are amino- and an imino-groups in 2-APC molecules which are responsible for ESIPT units. In the ethanol, 2-APCs can associate with ethanol molecules through hydrogen bonding to form six-membered complexes, enabling them to undergo an efficient ESIPT process and displaying strong ESIPT fluorescence. In the presence of Fe^3+^, the coordination of the amino- and an imino-group in 2-APC with Fe^3+^ may suppress the ESIPT reaction, resulting in the fluorescence quenching. These results agree with the FL spectra, which clearly verified our proposed detection principle.

### 2.5. Selectivity of 2-APC Probe to Fe^3+^

We then evaluated the response behavior of 2-APC toward Fe^3+^, the other metal ions and common anions in the ethanol–HEPES buffer system with relative fluorescence intensity. Upon the addition of biologically active cations, viz., Na^+^, K^+^, Ca^2+^, Mg^2+^, Mn^2+^, Zn^2+^, Cd^2+^, Cr^3+^, Co^2+^, Ni^2+^, Pd^2+^, Hg^2+^, Cu^2+^ and Pb^2+^, as well as the anions Cl^−^, SO_4_^2−^, CO_3_^2−^ and HPO_4_^2−^, the 2-APC probe does not show significant change in the emission pattern and intensity upon excitation at 345 nm. In the presence of Fe^3+^ in the ethanol–HEPES mixed solution (10 mM, pH 6.5) of 2-APC, the green fluorescence is obviously weak (Figure 4). The fluorescence performance of 2-APC toward Fe^3+^ in the presence of metal ions and common anions was also investigated.

The other metal ions and common anions did not affect the emission pattern and intensity of 2-APC (data not shown). These results indicated that the 2-APC exhibited good selectivity toward Fe^3+^ in the ethanol–HEPES buffer system and has potential for monitoring Fe^3+^ in a biological environment without severe interference from other biologically relevant species.

### 2.6. Detection of Fe^3+^ Using 2-APC as a Probe

Considering the sensitive response of 2-APC dyes to Fe^3+^, we build a fluorescence method for Fe^3+^ detection using 2-APC as a probe. As shown in Figure 5, the emission decreases gradually with the increase in Fe^3+^ concentrations, which is linearly correlated ranging from 0 to 13 μM L^−1^. The detection limit based on the formation of 2-APC-Fe^3+^ complexes was also evaluated at 7.5 nM L^−1^ (3σ). Obviously, these results confirmed that 2-APC has remarkably high sensitivity towards Fe^3+^ ions.

To examine the applicability of the proposed probe in practical samples, the probe was employed to detect Fe^3+^ in water and vegetables under the optimized conditions (data not presented). The river water and vegetable samples were prepared as described in the Experimental Section. The water and vegetable samples were spiked with standard Fe^3+^ solutions at different concentration levels and then analyzed using the 2-APC probe method. The results are given in Table 2. One can see that the recovery study of spiked Fe^3+^ measured by the probe shows satisfactory results (97.2–103.4%), with the reasonable RSD ranging from 3.1 to 3.8%. The proposed method seems useful for the determination of Fe^3+^ in real samples.

### 2.7. Cell-Imaging Using 2-APC

The cytotoxicity assay results showed that 2-APC less than 150 μM had no obvious toxicity to cells. In this case, we examined the feasibility of 2-APC for cell-imaging using A549 cells as model cell lines. The fluorescent images of A549 cells incubated with 2-APC and Fe^3+^+ 2-APC were shown in Figure 6. The cells treated with only the 2-APC probe for 1 h at 37 °C display intracellular light red fluorescence (Figure 6A). Meanwhile, the same cells with a pre-incubation in the medium containing Fe^3+^ and the subsequent load of 2-APC show relatively weak red fluorescence (Figure 6D).

The bioimaging results indicate that the 2-APC exhibits good cell membrane permeable and no apparent toxicity to the A549 cells, thus, the 2-APC displayed low toxicity toward the cultured cell lines under the experimental conditions. A549 can also track Fe^3+^ in living cells. In addition, it can be seen that the ingested 2-APC dye exhibited good color rendering and mainly concentrated in the cytoplasm.

## 3. Materials and Methods

### 3.1. Reagents

N-ethyl-3-carbazolecarboxaldehyde, malononitrile, acetophenone and ammonium acetate (NH4Ac) were purchased from Sinopharm Chemical Reagent Co., Ltd. (Shanghai, China). All chemicals and solvents were used as received without further purification. The stock solution of 10.0 mM/L Fe^3+^ was prepared by dissolving ferric sulfate in a HEPES buffer (10 mM, pH 6.5). The stock solution of 2-APC (10 mM/L) was obtained by dissolving the 2-APC in a little DMSO and setting the volume with HEPES buffer (10 mM, pH 6.5). Working solutions were prepared by diluting the corresponding stock solutions to an appropriate volume with the same PBS buffer whenever required. The human lung adenocarcinoma cell line A549 (A 594 cells) was purchased from Shanghai Institutes for Biological Sciences (Shanghai, China).

### 3.2. Instruments

The fluorescence measurements were carried out on an F-7000 fluorescence spectrophotometer (Hitachi, Tokyo, Japan). UV-vis absorption spectra were recorded on a Cary 60 UV-vis spectrophotometer (Agilent Technologies, Mulgrave, VIC, Australia). 1HNMR spectra were acquired with a Bruker AVB-400 MHz NMR spectrometer (Bruker BioSpin, Fällanden, Switzerland). Fourier transform infrared (FT-IR) spectra in KBr were recorded with a WQF-510 FT-IR spectrometer (Beijing Rayleigh Analytical Instrument Co., Ltd., Beijing, China). All photographs were obtained using Invitrogen EVOS M7000 from Thermo Fisher Scientific (Waltham, MA, USA).

### 3.3. Synthesis of 2-APC

The compounds 2-amino-4-(9-ethyl-9H-carbazol-3-yl)-6-phenylnicotinonitrile (2-APC) were synthesized by “four-component one-pot” reaction according to reported procedures [40] with minor modification. Briefly, 25 mL ethanol was added with N-ethyl-3-carbazolecarboxaldehyde (2.23 g, 0.01 mol), malononitrile (0.66 g, 0.01 mol), ammonium acetate (3.1 g, 0.04 mol) and acetophenone (1.2, 0.01 mol). The mixture was refluxed in an oil bath for 4 h. After being cooled to room temperature, the reaction mixture was extracted with 120 mL EtOAc. The organic phase was washed with saturated aqueous NH_4_Cl (50 mL × 3), water and brine, dried over anhydrous Na_2_SO_4_ and concentrated in vacuum. The resulting residue was then purified by flash chromatography on silica gel to afford the desired compounds. Light brown solid (2.8 g, 72%); m. p.: 197–198 °C. HRMS-EI: m/z [M+] calcd. for C26H20N4: 388.46; found:388.17. ^1^H NMR (400 MHz, DMSO) δ 8.51, 8.33, 8.17, 8.10, 8.02, 7.90, 7.84, 7.72, 7.58, 7.56, 7.54, 7.48, 7.33, 6.74, 4.51, 1.35; ^13^C NMR (101 MHz, DMSO) δ 162.04, 156.41, 154.24, 143.43, 140.89, 129.89, 129.22, 128.41, 128.26, 127.78, 127.31, 126.01, 123.11, 121.93, 121.46, 120.30, 118.95, 116.10, 114.78, 113.82, 111.91, 110.84, 103.36, 85.41, 40.39, 14.24. IR (KBr): 3447.01, 1630.25, 1471.03, 1490.49, 1381.31, 1336.64, 1223.75, 1154.71, 1154.96, 1120.56, 798.13, 748.59.

### 3.4. Analytical Procedure

First, a 0.5 mL 2-APC stock solution (10 mM/L) was diluted into 5 mL with HEPES buffer (10 mM, pH 6.5) in a 10 mL graduated tube. Next, a certain amount of Fe^3+^ standard solution (or sample solution) was added to the test tube. After using HEPES buffer for cooling capacity to 10 mL, the mixture was shaken thoroughly and left to react for 10 min at room temperature prior to the fluorescence measurements. Simultaneously, reagent blanks were acquired without Fe^3+^ standard solution or sample solutions. Final, the fluorescence intensity of the emission peaks at 575 nm for the test solution and the reagent blank (F_575_) were directly recorded on an F-7000 fluorescence spectrophotometer with an excitation wavelength of 345 nm.

### 3.5. Preparation of Real Samples

The river water was obtained from Xiangjiang River (Changsha, China), and the tap-water was collected from the laboratory of Changsha University of Science and Technology. For the sample preparation, the water samples (100 mL, three parallel samples) were filtrated through a filter paper (28 cm), and the filtrate was then stored at 4 °C. The cabbage was obtained from the local supermarket in Changsha (China). The vegetable samples (10 g) were cleaned with pure water and followed by crushing the rinsed samples and extracting with 50 mL HEPES buffer (10 mM, pH 6.5). After filtration, the filtrate samples were diluted with HEPES and used for determination.

### 3.6. Cytotoxicity Assays

A549 cells were used as the model cell line to evaluate the in vitro cytotoxicity of 2-APC fluorescent dyes using the MTT assay. Firstly, the A549 cells were incubated in a 96-well plate (5000 cells/well), which was added with a 100 μL DMEM solution containing 7 % fetal bovine serum (FBS) under humidified 5% CO_2_ atmosphere at 37 °C. After a 24 h-incubation, the medium was replaced, and different concentrations of 2-APC were added for another day-incubation. Next, 50 μL test solution containing 10 mg mL^−1^ MTT reagents was added into each well, further incubated at 37 °C for additional 4 h. Finally, the medium mass was removed, and 150 μL DMSO was then added and shaken for 15 min in the dark. The resulting mixtures were employed for UV-vis spectrometry by a microplate reader at the wavelength of 490 nm.

### 3.7. Fluorescence Imaging of Living Cells

Intracellular distribution of 2-APC against A549 cells and the fluorescence imaging were investigated using a laser scanning confocal microscope (CLSM). Typically, A549 cells were placed in a 35 mm glass-bottom culture dish at a density of 1 × 10^4^ cells per dish, letting it grow overnight in a 2 mL DMEM with 10% fetal bovine serum with 5% CO_2_ humidified atmosphere at 37 °C for 36 h. After that, the original medium was eliminated, and a 2 mL fresh medium containing 2-APC (1.5 mg/L) was then added into each dish. Followed by incubation for 12 h, ice-cold PBS buffer (20 mM, pH 7.4) was added to rinse the remaining 2-APC. After washing with an ice-cold PBS buffer three times, the resulting cells were then further observed under CLSM.

## 4. Conclusions

In summary, we developed a new-type ESIPT fluorophore with tricolor-emissive properties that exhibit normal emission, ESIPT-emission and aggregation-induced fluorescence. In aprotic solvents, blue fluorescence was clearly observed, which is assigned to the normal emission of 2-APC. Meanwhile, in protonic solvents (ethanol, water), the longer wavelength fluorescence emerges due to the partnered excited-state intermolecular proton transfer (ESIPT) process. Interestingly, 2-APC can also show aggregation-induced emission in poor solvents or solid-state, eliciting a red fluorescence. The theoretical evaluation using the Extended Hückel method demonstrated that the installation of the cyano-group on pyridine ring can increase both the acidity of D and the alkalinity of A, enabling the D-A system to undergo proton transfer smoothly. The ESIPT reaction of 2-APC can be eliminated by Fe^3+^ and cause fluorescence quenching, which makes it an outstanding fluorescent probe. From the results, it can be concluded that the probe 2-APC is highly selective and sensitive towards detecting Fe^3+^ and cell-imaging. We believed that the demonstrated strategy based on “four-component one-pot” synthesis can pioneer a new route to achieve proton solvent mediated-ESIPT fluorophores with multi-mode emissions.

## Data Availability

Not applicable.

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
