# Peer review of "A Solvent-Mediated Excited-State Intermolecular Proton Transfer Fluorescent Probe for Fe3+ Sensing and Cell Imaging"

_molecules, 2022, doi:10.3390/molecules27020516_

Round 1
Reviewer 1 Report
Authors have synthesized a new type excited-state intermolecular proton transfer (ESIPT) fluorophor (2-APC) using four-component one-pot reaction for Fe3+ sensing and cell imaging. The synthesized 2-APC was characterized and evaluated for Fe3+ sensing and cell imaging. The proposed work is interesting for publication in Molecules. However, the manuscript needs some revisions which are suggested below:
Title of article: ESIPT is not a standard abbreviated form. Authors are advised to expand this abbreviation in the title of the paper for better understanding to the readers.
Abstract: Authors are advised to include more quantitative information in order to enhance the readability of the article.
Introduction: The rationale and objective of studies should be clearly indicated in introduction section.
Figure 1: The latters in Figure are not readable. Authors are advised to increase the size of the latters and improve the resolution of the Figure.
Figure 3: The latters in Figure are not readable. Authors are advised to increase the size of the latters and improve the resolution of the Figure.
All the abbreviations, acronyms, and initials must be defined under abstract, text, table, and figure in their first appearance.
Scheme 1: The latters in Scheme are not readable. Authors are advised to increase the size of the latters and improve the resolution of the Scheme.
Figure 5: The latters in Figure are not readable. Authors are advised to increase the size of the latters and improve the resolution of the Figure.
Table 2: The term [a] should be in superscript. It should also be included in table caption or table.
Reference list: kindly update the reference list with latest articles of 2020 and 2021.
Author Response
Response to reviewer 1
- Comment: ESIPT is not a standard abbreviated form. Authors are advised to expand this abbreviation in the title of the paper for better understanding to the readers.
Response: Thank you very much for your professional comment. We expanded this abbreviation in the title of the revised manuscript.
- Comment: Authors are advised to include more quantitative information in order to enhance the readability of the article.
Response: Thank you for the suggestion. According to the suggestion, we added more quantitative information in the revised paper.
- Comment: The rationale and objective of studies should be clearly indicated in introduction section.
Response: We added the rationale and objective of studies in the revised paper.
- Comment: Figure 1: The latters in Figure are not readable. Authors are advised to increase the size of the latters and improve the resolution of the Figure.
Response: Figure 1 was enlarged in the revised paper.
- Comment: Figure 3: The latters in Figure are not readable. Authors are advised to increase the size of the latters and improve the resolution of the Figure.
Response: Figure 3 was enlarged in the revised paper.
- Comment: All the abbreviations, acronyms, and initials must be defined under abstract, text, table, and figure in their first appearance.
Response: We corrected the abbreviations, acronyms, and initials in the corresponding position in the revised paper.
- Comment: Scheme 1: The latters in Scheme are not readable. Authors are advised to increase the size of the latters and improve the resolution of the Scheme.
Response: Scheme 1 was enlarged in the revised paper.
- Comment: Figure 5: The latters in Figure are not readable. Authors are advised to increase the size of the latters and improve the resolution of the Figure.
Response: Figure 5 was enlarged in the revised paper.
- Comment: Table 2: The term [a] should be in superscript. It should also be included in table caption or table.
Response: According to the suggestion, the term [a] was corrected in the revised paper.
- Comment: Reference list: kindly update the reference list with latest articles of 2020 and 2021.
Response: According to the suggestion, we added latest article of 2020 and 2021.

Reviewer 2 Report
The authors report on the development of a new molecular probe that has been designed to sense Fe3+ ions. The work is novel but there are a few minor issues that the authors should consider addressing prior to publication.
Minor comments:
- The authors mention FTIR spectra for the compounds but this could not be found in SI. Furthermore, the resolution of NMR spectra should be enhanced and this should be added to the main manuscript, along with FTIR spectra, in my opinion.
- Some discussion relating to the further work that is required for optimization of these probes, plus the application of these molecules would be useful in the R&D section.
- The authors report in the Methods section that a cytotoxicity study was performed using the MTT assay. However, I cannot see any data in the manuscript relating to this assay. Can the authors please include this data and indicate the concentrations by which these probes can be used for A549 cellular assays?
- The font size in most figures is quite small and could be increased.
- Scale bars are required for FIg 2B and Fig 6.
- Line 12 abstract should be: "By the installing of a cyano-group..."
Author Response
Response to reviewer 2
- Comment: The authors mention FTIR spectra for the compounds but this could not be found in SI. Furthermore, the resolution of NMR spectra should be enhanced and this should be added to the main manuscript, along with FTIR spectra, in my opinion.
Response: Thank you very much for your professional comment. We presented the FTIR spectrum for the compound 2-APC in the revised Supporting Materials.
- Comment: Some discussion relating to the further work that is required for optimization of these probes, plus the application of these molecules would be useful in the R&D section.
Response: Thank you for the suggestion. The data for optimization of the probe was not presented in the paper.
- Comment: The authors report in the Methods section that a cytotoxicity study was performed using the MTT assay. However, I cannot see any data in the manuscript relating to this assay. Can the authors please include this data and indicate the concentrations by which these probes can be used for A549 cellular assays?
Response: We added the data and indicate the concentrations by which these probes can be used for A549 cellular assays in the revised paper.
- Comment: The font size in most figures is quite small and could be increased.
Response: Figures were enlarged in the revised paper.
- Comment: Scale bars are required for FIg 2B and Fig 6.
Response: Scale bars were presented in the revised paper.
- Comment: Line 12 abstract should be: "By the installing of a cyano-group..."
Response: "By the installing of a cyano-group..." was corrected in the revised paper.